# Regular group exercise is associated with improved mood but not quality of life following stroke

Michelle N. McDonnell[1], Shylie F. Mackintosh[1], Susan L. Hillier[1] and Janet Bryan[2]

[1] International Centre for Allied Health Evidence, University of South Australia, Adelaide, South Australia, Australia
[2] School of Psychology, Social Work and Social Policy, University of South Australia, Australia

## ABSTRACT

**Purpose.** People with stroke living in the community have an increased prevalence of depression and lower quality of life than healthy older adults. This cross-sectional observational study investigated whether participation in regular exercise was associated with improved mood and quality of life.

**Methods.** We recruited three groups of community dwelling participants: 13 healthy older adults, 17 adults post-stroke who regularly participated in group exercise at a community fitness facility and 10 adults post-stroke who did not regularly exercise. We measured mood using the Depression, Anxiety, Stress Scale (DASS) and quality of life using the Assessment of Quality of Life (AQoL) scale.

**Results.** Levels of stress and depression were significantly greater in the people with stroke who did not undertake regular exercise ($p = 0.004$ and $p = 0.004$ respectively), although this group had more recent strokes ($p < 0.001$). Both stroke groups had lower quality of life scores ($p = 0.04$) than the healthy adults.

**Conclusions.** This small, community-based study confirms that people following stroke report poorer quality of life than stroke-free individuals. However, those who exercise regularly have significantly lower stress and depression compared to stroke survivors who do not. Future research should focus on the precise type and amount of exercise capable of improving mood following stroke.

## INTRODUCTION

Mood disorders have been recognized as a complication following stroke since 1977 (*Folstein, Maiberger & McHugh, 1977*). Approximately one-third of stroke survivors have mood disorders, with depression and anxiety most frequently measured (*Lees et al., 2012*). Carers of stroke survivors report disturbances in mood as the most stressful stroke-related problem (*Haley et al., 2009*) and despite recommendations published in international guidelines to routinely assess mood post-stroke (*Lindsay et al., 2010*; *National Stroke Foundation, 2010a*), evidence-based interventions for mood disturbances are rarely provided (*National Stroke Foundation, 2010b*).

Corresponding author
Michelle N. McDonnell,
michelle.mcdonnell@unisa.edu.au

Exercise programs can improve mood through reducing depression and anxiety in healthy adults (*Byrne & Byrne, 1993*; *Howard et al., 2013*) and there are pilot data to suggest that a cardiac rehabilitation program for secondary stroke prevention may also decrease depression (*Lennon et al., 2008*). We investigated the benefits of regular exercise on mood and quality of life in a community setting involving three different groups—those currently exercising who had experienced a stroke, healthy adults without stroke and people with stroke referred to the community centre but not currently exercising.

## METHODS

### Participants

Participants were recruited from the Centre for Physical Activity in Ageing (CPAA), a unit of a large metropolitan hospital. People living in the surrounding community may be referred to the centre by their medical practitioner or following inpatient or community rehabilitation. Healthy older adults exercising at least once a week (HE) and people with stroke exercising at least once a week (SE) were recruited via flyers posted at the Centre. The third group was recruited via referrals to the centre for people post-stroke who wished to commence a fitness program but had not commenced exercise classes (STR).

As this was a cross-sectional observational study with a sample of convenience, participants were recruited to a group based on their characteristics after which we assessed mood and quality of life. Participants were excluded if they were diagnosed with dementia or if they had insufficient communication skills to complete the questionnaires. There were no age restrictions. This study was approved by relevant ethics committees and participants provided written informed consent in accordance with the Declaration of Helsinki.

### Assessment tools

The Montreal Cognitive Assessment (MoCA) was used to screen for mild cognitive impairment (MCI), with a cut-off score of 26 or above indicating normal range (*Nasreddine et al., 2005*). The Functional Ambulation Category (FAC) was used to classify the ability to ambulate with or without assistance (*Mehrholz et al., 2007*).

Mood disorders were measured using the Depression, Anxiety, Stress Scale (DASS) (*Crawford & Henry, 2003*) and quality of life using the Assessment of Quality of Life (AQoL) (*Hawthorne, 2000*). The DASS measures the three aspects of mood on a Likert scale allowing a maximal score of 42 for each of the three subscales. The DASS exhibits very good reliability (Cronbach's $\alpha = 0.90$ for the anxiety scale, 0.95 for the depression scale and 0.93 for the stress scale score) and internal and convergent validity is high for both depression and anxiety (with the Personal Disturbance Scale, $r = 0.78$ and $r = 0.72$, respectively) (*Crawford & Henry, 2003*). The AQoL was used to assess health-related quality of life in five dimensions: independent living, social relationships, physical senses, psychological well-being, and illness. It has been validated for use in people with stroke (*Sturm et al., 2002*) and scores range from 15 to 60, with a higher score indicating poorer quality of life. The AQoL also has sound psychometric properties, with good internal consistency (Cronbach's alpha $\alpha = 0.87$) and test-retest reliability ($r = 0.8$) (*Hawthorne, 2000*).

## Intervention

The Centre for Physical Activity in Ageing conducts approximately 40 circuit-style classes per week, with 6 dedicated stroke fitness classes and several "slow circuit" classes which are appropriate for people with impaired mobility. People referred to the centre from the community or following rehabilitation are provided with an initial 12 week program which is then reviewed by an exercise physiologist and continued if appropriate. Participants in the exercise groups (HE and SE) attended low to moderate intensity circuit classes, with approximately 20 participants in a class, conducted by a trained fitness instructor. The 60 min class started with a group warm up (standing or seated if necessary) for 5 min and cool down at the end of the class for 5 min. For the remaining 50 min each participant had their own tailored program devised by an exercise physiologist, according to their needs, and supervised by a fitness instructor with a carer present if required to assist with accessing the equipment. A typical class involved strengthening exercises with hydraulic equipment, aerobic exercise with a bike, treadmill or arm ergometer, walking and other weight bearing exercises in the parallel bars, and other higher level activities such as stair climbing or rowing machines. All participants in the SE and HE groups took part in at least one session per week.

## Statistical analysis

Data were inspected for normality and transformed appropriately if required. Group differences were compared using one way analysis of variance (ANOVA; SPSS Version 18) for age, years of education, FAC, MoCA, exercise history, years since stroke, depression, anxiety and stress, and AQoL, and a Chi-square test for group differences in sex. Levine's test assessed homogeneity of variance for each group. In cases of unequal variances, analysis was repeated using the Welch Robust Test of Equality of Means. Post-hoc analyses following ANOVA were preplanned to assess for differences between groups using Tukey's HSD test or Tamhane's T2 post hoc test (for unequal variances) to reduce the risk of a Type 1 error for multiple comparisons. Results were considered significant with $p < 0.05$.

## RESULTS

There were 40 participants; 17 people post-stroke undertaking exercise (SE), 13 healthy exercisers (HE) and 10 participants post-stroke who were not exercising (STR). Participant characteristics are shown in Table 1. As this study involved a single assessment session, no participants dropped out of the study.

There were no significant differences in age between groups (ANOVA, $F_{(2,37)} = 0.791$, $p = 0.46$) although there was a non-significant trend towards more females in the HE group (9/13) compared to the stroke groups (5/17 and 2/10) (Chi-square test, Pearson $X^2 = 5.632$, $p = 0.06$). STR participants had significantly fewer years of education (ANOVA, $F_{(2,37)} = 1.627$, Welch $t$ statistic $= 3.899$, $p = 0.04$) than both other groups and on average had suffered a stroke more recently than the SE participants (Mann–Whitney Test, $U = 17$, $p < 0.001$).

All participants in the HE and STR groups scored a 5 on the FAC, indicating that they were able to walk independently anywhere, including stairs (with a walking aid

**Table 1  Participant characteristics for the three groups with mean scores and standard deviations.**

| | Stroke and Exercise (SE) $n = 17$ | Healthy Exercisers (HE) $n = 13$ | Stroke no Exercise (STR) $n = 10$ |
|---|---|---|---|
| Age (years) | $70 \pm 10$ | $69 \pm 7$ | $65 \pm 9$ |
| Males | 12 | 4 | 8 |
| Years of education | $11 \pm 3$ | $11 \pm 4$ | $9 \pm 1^*$ |
| Years since stroke | $8.9 \pm 6.9$ | N/A | $1.6 \pm 0.7^*$ |
| FAC category | $4.4 \pm 1.1^*$ | $5 \pm 0$ | $5 \pm 0$ |
| Exercise history (yrs) | $4.8 + 3.6$ | $6.8 + 4.3$ | N/A |
| MoCA score | $21.7 \pm 4.3^*$ | $25.8 \pm 2.1$ | $24.4 \pm 4.1$ |
| Left lesion | 8 | N/A | 5 |
| AQoL | $28.8 \pm 7.0$ | $23.3 \pm 6.0^*$ | $30.0 \pm 3.1$ |

**Notes.**

FAC, Functional Ambulation Category; MoCA, Montreal Cognitive Assessment; AQoL, Assessment of Quality of Life.

$^*$ $p < 0.05$.

if required). The SE group had significantly lower FAC scores (Kruskal-Wallis $H(2) = 7.504, p = 0.02$, Table 1) than both other groups. There was no significant difference in the number of years exercising for the SE and HE group (mean (SD) years exercising at least once per week for the SE group $= 4.1 \pm 2.6$, HE group $= 7.2 \pm 4.5$, Mann–Whitney test, $U = 3.844, p = 0.14$).

The MoCA demonstrated that the mean score for all three groups was below the cut off score of 26 indicating mild cognitive impairment. There was a significant main effect for group (ANOVA, $F_{(2,37)} = 4.721, p = 0.02$, Table 1) with the SE group score significantly lower than the HE group (SE group MoCA mean (SD) $= 21.7 \pm 4.3$, HE group MoCA $= 25.8 \pm 2.1$, Tukey test, $p = 0.1$); there was no difference between the SE and STR groups.

Both stroke groups reported significantly higher values on the AQoL indicating poorer quality of life (ANOVA, $F_{(2,37)} = 3.511, p = 0.04$) than the HE group as shown in Table 1. There was no difference between the STR and SE groups.

After logarithmic transformation as DASS scores were positively skewed, a one way analysis of variance of depression scores revealed a significant main effect for group (ANOVA, $F_{(2,37)} = 6.421, p = 0.004$) and post-hoc Tukey tests confirmed that the mean (standard error) scores for those in the STR group ($0.9 \pm 0.2$) were significantly greater than both SE ($0.4 \pm 0.1, p = 0.02$) and HE ($0.3 \pm 0.1, p = 0.004$) groups (Fig. 1). For stress there was also a significant main effect for group (ANOVA, $F_{(2,37)} = 6.367, p = 0.004$). Participants in the STR group reported significantly greater stress ($0.8 \pm 0.1$) than both the SE ($0.4 \pm 0.1$, Tukey test, $p = 0.02$) and HE groups ($0.3 \pm 0.1, p = 0.005$). The third subscale, anxiety, did not show a significant effect for group (ANOVA, $F_{(2,37)} = 2.004$, Tukey test, $p = 0.15$).

Regression analysis was considered to investigate patient characteristics which may impact upon exercise and mood. There was no significant correlation between depression/stress and age, sex, years since stroke, functional ambulation category, years of education or cognitive status. Further, there was no correlation between the MoCA and

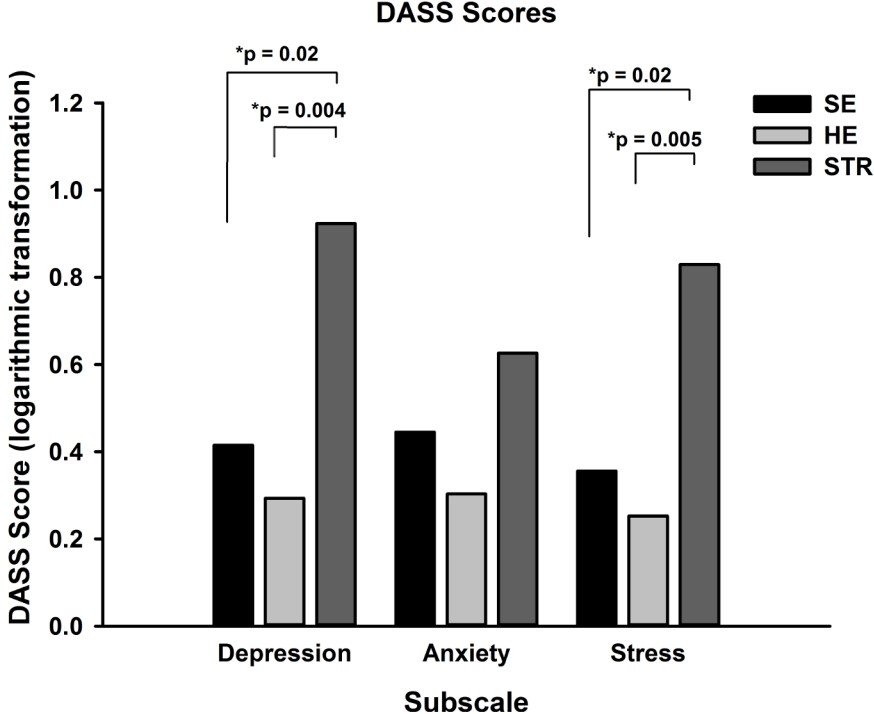

**Figure 1 DASS scores for the three groups.** Mean (SEM) DASS scores for each subscale were calculated and revealed that the STR group had significantly greater depression and stress than both HE and SE groups, with no difference between scores for anxiety.

quality of life or education. However, Pearson product-moment correlation coefficients revealed significant correlations were between DASS subscores and AQoL scores (depression $r = 0.453$, $p = 0.003$, stress $r = 0.387$, $p = 0.014$, anxiety $r = 0.572$, $p < 0.001$).

## DISCUSSION

This study indicates that those exercising in a group at least once per week reported significantly less depression and stress than people following stroke who were not currently exercising. While we cannot claim that exercise alone is responsible for this association, it is an intriguing possibility. The supervised exercise sessions were conducted in a group, so it could be that social contact and support may have contributed to lowered stress and depression (*Babyak et al., 2000*). In contrast, both stroke groups reported worse quality of life than the healthy exercisers which is likely reflecting their AQoL scores for dependence on medications, regular treatment by a doctor and often requiring assistance with activities of daily living.

There were higher scores for depression in the stroke group not taking part in exercise. Data regarding use of antidepressants were self-reported, with three participants in the STR group on a stable dose of antidepressants, two participants in the HE and none in the SE group. This suggests that although depression was more prevalent in the STR group, more individuals in this group were receiving treatment which strengthens our findings. In addition, difference in stress levels between groups also followed a similar pattern.

These results are based on a small sample, therefore further investigation of the potential for exercise to decrease post-stroke depression and stress are warranted.

Health-related quality of life was lower in both stroke groups. There was no difference between those who exercise and those who did not, suggesting that exercise is unlikely to mediate improved mood via improved quality of life. Quality of life, as measured with the AQoL tool reflects items such as dependence on others for assistance and on medications and regular medical treatment. Many stroke survivors are dependent on several prescription medicines for life, requiring close monitoring by their regular medical practitioner, so this is likely to have influenced AQoL scores. Further we have shown that quality of life correlates with depression, anxiety and stress scores, so even the stroke survivors who exercise have their quality of life influenced by psychological factors.

There were several baseline differences between groups which may impact upon our findings. Firstly, the SE group had significantly lower scores on the FAC, due to the inclusion of several individuals who needed assistance with their mobility. We may expect that those with greater mobility restrictions may have greater mood disturbance but this was not the case in this sample of stroke survivors. The STR group also had significantly fewer years of formal education but there was no evidence of lower cognitive scores in the STR group. Rather, the SE participants had significantly lower scores on the MoCA, suggesting more severe cognitive impairment. This is likely to be due to the duration of time since their stroke. These individuals were almost 10 years post-stroke and incidence of stroke is associated with an increased risk of dementia and cognitive decline (*Sachdev et al., 2004*). Finally, the STR group was assessed at a significantly shorter period following their stroke, on average 1.6 years post-stroke compared to 9 years in the SE group. This is unlikely to be a significant limitation as a recent multicentre trial provides evidence that depression and anxiety increased significantly 5 years after stroke compared with 6 months post-stroke (*Lincoln et al., 2012*) and a systematic review of observational studies indicated that the prevalence of depression was similar in acute, sub-acute and chronic phases after stroke (*Hackett et al., 2005*).

Another limitation of the study is the observational design; formal attendance was not recorded and participants were free to exercise *ad libitum*. However, the majority of exercisers did attend regularly, 1–2 times per week, with very few absences due to illness. It is also worth noting that these results are only generalizable to those individuals who chose to respond to the recruitment flyer; it is possible that those people following stroke and the healthy adults with lower mood states may not have chosen to take part in the study.

There is growing evidence to demonstrate that group exercise may influence mood following stroke. A pilot study comparing group exercise alone to group exercise and yoga found similar improvements in anxiety and depression between the two groups of stroke survivors (*Kahn et al., 2008*). Further, both a cardiac rehabilitation model of exercise (*Lennon et al., 2008*) or treadmill training (*Smith & Thompson, 2008*) improved self-reported depression in individuals post-stroke. However, another recent study suggested that combined aerobic and resistance exercise classes, similar to the present study, were not effective in improving depressive symptoms (*Hickey et al., 2012*). A large

randomized controlled trial recruiting stroke survivors and their families is underway to address this important area of research, although exercise is just one component of the planned intervention, along with cognitive behavioural therapy (*Gray et al., 2011*). While the current study is observational in design, the strength of this study is that we have compared mood states in community dwelling adults and people following stroke and can demonstrate the potential benefits of exercising in a group setting to improve mood, particularly stress and depression. This project provides preliminary data for proof of concept and provides justification for further research using more robust clinical designs to investigate the role that exercise has on mood post-stroke.

## CONCLUSION

This community based study provides some preliminary evidence that people post-stroke who exercise in a group setting regularly have significantly lower stress and depression compared to stroke survivors who do not exercise. Future research should focus on the precise type and amount of exercise capable of improving mood following stroke, the impact of exercise in the early versus late stage post-stroke and the specific contribution of group versus individual exercise programs.

### Funding

This work was funded in part by a Research Grant from the Centre for Metabolic Fitness, Australian Technology Network; the National Stroke Foundation, Australia; and the Division of Health Sciences, University of South Australia. MN McDonnell was supported by a National Health and Medical Research Council Research Training Fellowship. The funders had no role in study design, data collection and analysis, decision to publish, or preparation of the manuscript.

### Grant Disclosures

The following grant information was disclosed by the authors:
National Stroke Foundation Small Project Grant.
Centre for Metabolic Fitness, Australian Technology Network.
Division of Health Sciences, University of South Australia.
National Health and Medical Research Council Research Training Fellowship.

### Competing Interests

The authors declare there are no competing interests.

### Author Contributions

- Michelle N. McDonnell conceived and designed the experiments, performed the experiments, analyzed the data, contributed reagents/materials/analysis tools, wrote the paper, prepared figures and/or tables, reviewed drafts of the paper.
- Shylie F. Mackintosh conceived and designed the experiments, analyzed the data, reviewed drafts of the paper.

- Susan L. Hillier analyzed the data, reviewed drafts of the paper.
- Janet Bryan performed the experiments, analyzed the data, contributed reagents/materials/analysis tools, reviewed drafts of the paper.

## Human Ethics

The following information was supplied relating to ethical approvals (i.e., approving body and any reference numbers):

University of South Australia Human Research Ethics Committee, Protocol P090/09.

## Supplemental Information

Supplemental information for this article can be found online at http://dx.doi.org/10.7717/peerj.331.

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
