# Peer review of "Regular group exercise is associated with improved mood but not quality of life following stroke"

_PeerJ, doi:10.7717/peerj.331_

## Round 0.1 · original submission · Minor Revisions

Dear Authors, The two peer reviewers have given very important suggestions to improve the quality of the manuscript.Please do the appropriate revisions so as the resubmitted revised manuscript can be sent to the same peer reviewers.

·

Basic reporting

I think this paper is nice and clear, there are many ways to extend this research further. It would be nice to have the participants more closely matched, so more recent stroke patients undergoing exercise so then it is easier to compare across the groups

Experimental design

I would like a little bit more details on the type of exercise participants where doing as it's not very clear

Validity of the findings

The paper is missing f & t values. In line 87, the f value and degrees of freedom are missing from the reported ANOVA results. In line 89 the chi-square value is missing. In lines 90 & 91, the t value and degrees of freedom are needed. In lines 94, 95, 97, 98, 101, 105, 107, 108, 110 & 111 the t/f values are missing as well as the degrees of freedom.
I think it would be worth correlating the DASS scores with MoCA, time of stroke,years of education, FAC & AQoL. I would be a little wary of the fact the mean MoCA scores are all below the 26 cut off point. I would enter those and year of stroke as a covariate just to check whether play a role in the results.

·

Basic reporting

This is an easy-to-follow piece of academic writing. The reporting has been done clearly and meets the standards of the journal. Topic has been introduced sufficiently and can guide readers on what to expect from the rest of the work.

Experimental design

Research objectives/questions can be clearly understood. The significance of the research is noted. Research findings may however raise some issues due to its observational design. Authors, however, have given sufficient justifications. One recommendation is to be more stringent on the inclusion criteria for the participants. Since exercise is an important variable in this study, it has to be controlled more systematically. This can be done by including only stroke survivors who have been doing exercise for certain period of time (SE) vs. those who have not commenced the exercise (STR). The duration or the period in which they have been exercising has to be a criteria to choose participants in SE. Authors use the term ‘regular’ but there seems to be no evidence to show that SE is regular in their exercise. In other words SE exercises at least once a week and it is misleading to use ‘regular’ unless the time they have been doing this is stated. In other words, as of now readers have no clue for how long they have been active in the exercise.

Validity of the findings

The Result & Discussion section has been written clearly. Authors may consider to elaborate the reasons for the worse quality of life reported for the SE and STR (Line 118). Conclusions have been appropriately stated.

Additional comments

A nice piece and easy to follow research write-up. I support your view on the importance to have studies addressing the role of exercise in improving the mood of stroke survivors.

---

## Round 0.2 · accepted · Accept

Dear Authors,Congratulations for submitting the revised manuscript which is now clearer,more academic and suitable for publication.

Reviewer 1 ·

Basic reporting

the necessary changes have been made now

Experimental design

Much clearer with the added exercise information

Validity of the findings

results are clear

Additional comments

addressed the issues raised